# Development of Heterojunction c-Si/a-Si_1−x_C_x_:H PIN Light-Emitting Diodes

**DOI:** 10.3390/mi13111948

**Published:** 2022-11-10

**Authors:** Maricela Meneses-Meneses, Mario Moreno-Moreno, Alfredo Morales-Sánchez, Arturo Ponce-Pedraza, Javier Flores-Méndez, Julio César Mendoza-Cervantes, Liliana Palacios-Huerta

**Affiliations:** 1National Institute of Astrophysics, Optics and Electronics, INAOE, Electronics Group, Tonantzintla, Puebla 72840, Mexico; 2Department of Physics and Astronomy, University of Texas at San Antonio, One UTSA Circle, San Antonio, TX 78249, USA; 3Facultad de Electrónica, Meritorious Autonomous University of Puebla, Benemérita Universidad Autónoma de Puebla, Puebla 72000, Mexico

**Keywords:** amorphous silicon-carbide, photoluminescence, electroluminescence, light-emitting diode

## Abstract

In this work, we explored the feasibility of the fabrication of PIN light-emitting diodes (LEDs) consisting of heterojunctions of amorphous silicon-carbide (a-Si_1−x_C_x_:H) thin films and crystalline silicon wafers (c-Si). The objective is the future development of electro-photonic systems in the same c-Si wafer, containing transistors, sensors, LEDs and waveguides. Two different heterojunction LEDs were fabricated consisting of PIN and PIN^+^N structures, where a-Si_1−x_C_x_:H thin films were used as P-type and I-type layers, while an N-type c-Si substrate was used as an active part of the device. The amorphous layers were deposited by the plasma-enhanced chemical vapor deposition (PECVD) technique at a substrate temperature of 200 °C. The PIN device presented electroluminescence (EL) only in the forward bias, while the PIN^+^N device presented in both the forward and reverse biases. The EL in reverse bias was possible due to the addition of an N^+^-type a-Si:H layer between the c-Si substrate and the I-type a-Si_1−x_C_x_:H active layer. Likewise, the EL intensity of the PIN^+^N structure was higher than that of the PIN device in forward bias, indicating that the addition of the N-type a-Si:H layer makes electrons flow more efficiently to the I layer. In addition, both devices presented red EL in the full area, which is observed with the naked eye.

## 1. Introduction

The development of light-emitting diodes (LEDs) based on silicon (Si) has received significant attention due to their compatibility with the Complementary Metal-Oxide-Semiconductor (CMOS) technology and the feasibility of integrating into the same chip electronic devices (transistors, diodes, sensors and waveguides) with the objective to develop monolithic photonics systems [1]. To achieve the above, it is important to have both compatibility of materials and low-temperature processing, where LEDs can be fabricated in a post-process on Si wafers already containing CMOS circuits and sensors.

Silicon-based materials, such as silicon carbide (SiC), silicon oxide (SiO_X_), silicon nitride (SiN_X_) and silicon-rich oxide (SRO), are the main candidates for the development of light-emitting devices because of their luminescent properties [2,3,4,5]. Particularly, the amorphous phase of silicon carbide (a-Si_1−x_C_x_:H) has certain characteristics that make it more attractive for LEDs than the other mentioned materials, such as its high intensity of photoluminescence (PL) as deposited (without any thermal treatment), lower optical energy gap (E_opt_) [6], and the fact that it can be synthesized at lower deposition temperatures (<200 °C) [7], allowing for its application to be extended to flexible substrates, glasses and polymers. In addition, it is possible to adjust its optical, electrical and structural properties by modifying its chemical composition [8,9,10,11,12].

The first observation of visible photoluminescence on an amorphous semiconductor was made on hydrogenated amorphous silicon (a-Si:H), prepared by glow discharge decomposition of silane-ethylene mixtures [9], and the first visible electroluminescence was observed in an amorphous hydrogenated silicon-carbon (a-SiC:H) alloy in 1983 [10]. Thanks to the above, a visible light-emitting PIN diode based on a-SiC:H was developed in 1985, and it was capable of emitting light in red, orange, yellow and green regions of the visible spectra by varying the optical energy gap (E_opt_) of the active layer [11]. The emitting light in different colors has also been controlled by adjusting the E_opt_ of the I-layer and the carrier injector P-layer [12].

It has been reported that the electroluminescence (EL) peak energy for PIN structures based on a-SiC:H active layer is lower than its respective PL energy, and the EL peak is usually at the red region for E_opt_ values near 2.6 eV [11,13,14,15]. The visible EL in the red region has also been observed in an a-SiC:H diode in reverse bias (RB) at −15 V. The reverse current producing the radiative emission was mainly related to a field-enhanced thermal injection mechanism, while current-voltage characteristics show that the carriers are introduced mainly by a tunneling mechanism [16].

Most of the research work performed until now on a-SiC: H-based LEDs has been developed on glass substrates; as well, to increase the efficiency of these PIN devices, complex structures (five or more layers) have been used. On the other hand, silicon wafer-based LEDs have been fabricated using high-temperature processes [6,17] or electrochemical etching [18]. Nevertheless, processes involving high temperatures (up to 1100 °C) limit the compatibility with the Si CMOS-based technology [19]. 

The Si CMOS technology established at present is based on P-type Si wafers (for N-channel MOS transistors and N-type wells for the P-channel MOS transistors). Therefore, the research and development on the use of PIN LEDs using N-type wells as an active part of the light-emitting devices on the wafer surface are important. 

In this work, we have explored the feasibility of PIN LED fabrication for their future incorporation in Si wafers with CMOS circuits. Two different PIN and PIN^+^N light-emitting devices were fabricated on N-type c-Si substrates at low temperatures without any thermal treatment, using a-Si_1−x_C_x_:H active layers. We have characterized the devices, optically and electrically, particularly the electroluminescence, in forward and reverse bias.

## 2. Experimental Details

The devices were fabricated on N-type Si-wafers (100) with a resistivity of 3–10 Ω·cm. A standard RCA cleaning process cleaned these wafers. Then, a 500 nm thick aluminum layer was deposited by an electron-beam evaporator from Balzers on the back side of the Si wafers and annealed in forming gas at 420 °C for 20 min to form an ohmic contact. The N-type substrate acts as an active part of the LED. Each amorphous layer was prepared independently at a substrate temperature of 200 °C in a Plasma-Enhanced Chemical Vapor Deposition (PECVD) multichamber system from MVSystems working at a Radio Frequency (RF) of 13.56 MHz, which contains three chambers for I, P and N thin films deposition. The I-layer (a-Si_1−x_C_x_:H) was deposited on the front side of the Si substrate, followed by the P-type layer consisting of a boron-doped (B_2_H_6_) a-Si_1−x_C_x_:H layer (a-SiC:H, B). Finally, a 100 nm thick layer of indium thin oxide (ITO) was deposited in a sputtering system, model ATC Orion from AJA, on the top of the P-type layer and a photolithographic step was performed to define devices areas of 1 mm^2^. 

The PIN^+^N structure was fabricated similarly to the previous structure but with an addition of a phosphorous (PH_3_) doped amorphous silicon layer (N-type a-Si:H) between the N-type Si substrate and the I-layer. A schematic diagram of both structures is shown in Figure 1. The deposition parameters of the films are shown in Table 1. It is worth mentioning that the electrical and optical properties of the N^+^ (a-Si:H, P), P (a-SiC:H, B) and I (a-Si_1−x_C_x_:H) layers were optimized previously [20,21].

The E_opt_ of P, I, and N+ layers were determined by Tauc’s plot, and their values are shown in Table 2. Additionally, this table shows the chemical composition of the active film, which is composed of silicon (Si), carbon (C) and oxygen (O) atoms. The photoluminescence of the active layer (a-Si_1−x_C_x_:H) at room temperature was measured using a Horiba Jobin Yvon Fluoro-Max3 spectrofluorometer with different excitation wavelengths from 254 nm to 540 nm (4.8–2.29 eV), and the PL signal was collected in the range of 400–900 nm. The current-voltage (I–V) characteristics of the PIN and PIN^+^N devices were measured using a Keithley 4200 SCS system, and the electroluminescence spectra were collected by means of an optical fiber connected to an Ocean Optics spectrometer model QE65000.

## 3. Results and Discussion

Physical properties such as thickness, E_opt_ and conductivity of the P, I, and N^+^ layers are shown in Table 2 and only for the a-Si_1−x_C_x_:H layer its chemical composition is shown. 

Figure 2 shows the normalized spectra of the PL intensity of a-Si_1−x_C_x_:H active layer with different excitation photon energy from 254 nm (4.8 eV) to 500 nm (2.48 eV). We can see that as the excitation energy decreases, the PL emission spectrum is blue-shifted and then red-shifted because of the activation of different recombination centers. PL is attributed to radiative recombination between an electron and a hole through defects and radiative recombination from the conduction band tail to the valence band tail [22,23,24].

Figure 3 shows the current density—voltage (J–V) characteristics in forward bias (FB) and reverse bias (RB) of the PIN and PIN^+^N devices, where it can be seen that, in FB, at voltages larger than 17 V, the current density grows exponentially for both kinds of devices, as expected. Since the barrier height at the N/I interface of the PIN device is higher than that of the PIN^+^N device (at the N^+^/I interface), then there is no current flow in the RB of the PIN device, which is typical behavior in this kind of device. However, for the PIN^+^N device, the current density also grows exponentially, which can be due to the tunneling of the current through gradual potential barriers at the N/N^+^/I and I/P interfaces [25].

The visible light emission of the PIN device was only observed in FB at voltages higher than 20 V, while for the PIN^+^N device, visible light emission was observed under FB and RB for voltages larger than 20 V and close to −25 V, respectively. The difference in the electrical behavior of the PIN^+^N device with respect to the PIN device in RB is due to the additional N^+^ layer (a-Si:H). This layer reduces the barrier height of the carriers in the N/I interface (N-type a-Si:H/I a-Si_1−x_C_x_:H) and increases the carrier injection efficiency to improve EL intensity [26]. The reverse current producing the radiative emission is mainly attributed to a field-enhanced thermal injection mechanism where the carriers are mostly introduced by a tunneling mechanism [16].

Figure 4 shows PL spectra (with E_ex_ = 300 nm) of the a-Si_1−x_C_x_:H active layer and the EL spectra of the PIN (in FB) and the PIN^+^N (in FB and RB) devices at 26 V (and −26 V). The PL spectra are centered at 482 nm with a Full Width at Half Maximum (FWMH) of 120 nm. On the other hand, the maximum intensity of EL in FB was located at 770 ± 4 nm corresponding to the red emission with FWHM values of 240 nm and 200 nm for the PIN device and the PIN^+^N device, respectively. 

It is easy to see that there is a shift of 288 nm between the EL and PL spectra. Such broad spectra of light emissions could be due to the wider energy range of states located in the active layer [3,26]. The above shift has already been reported previously on SRO-based devices [27] and on SiC-based PN junctions (with a difference between EL and PL spectra of 75 nm) [28]. 

The shift between PL and EL can be caused by the distortion of lattices, the relaxation of carriers into deep gap states before the recombination takes place [3] and/or by the excitation of carriers by the electric field due to impact ionization in the localized states and in the extended states of the valence and conduction bands, resulting in the lower energy shift and broadening of the EL spectra compared to the PL spectra [10]. 

In Figure 2 and Figure 4, we can see that the light emission from the devices would be limited not only by the optical gap of the active layer but also by the excitation energy of electron-hole pairs; this is the injection energy level of electrons and holes from the P and N layers [13]. Both PIN and PIN^+^N devices emit at the same wavelength because they have the same active layer; their structure does not influence the emission color, so they have the same EL origin, which could be either by the recombination of two different defect states as tail states near the valence band edge of the I-layer or by recombination with free electrons via the tail states [13,29]. The light emitted by both devices was observed in the full area with the naked eye, as shown in Figure 5a,b. 

Figure 6a shows the EL intensity as a function of voltage (in absolute value) of the PIN device in FB and PIN^+^N device in FB and RB. In FB, the PIN device presented EL from 21 V to 26 V and the PIN^+^N device from 20 to 26, while in RB, only the PIN^+^N device showed EL from −25 V to −29 V.

As shown in Figure 6a, the EL intensity of the devices increases with the voltage, which indicates that electrons are efficiently injected into the a-Si_1−x_C_x_ film, which is more efficient in the PIN^+^N than in the PIN device due to the N-type a-Si:H film. The above is because a higher electric field on a-Si:H film induces the carriers to be injected into the active layer with a higher energy level; therefore, in the PIN^+^N structure, more electrons could be injected into the I-layer then recombine with holes to enhance the emission. Figure 6b shows the EL intensity as a function of current density. The PIN^+^N device presents a higher EL intensity at a lower current density compared to the PIN device.

The dominant conduction mechanism of our devices can be deduced from typical plots of current-voltage characteristics for each conduction mechanism, such as thermionic emission, Fowler–Nordheim tunneling or Space Charge Limited Conduction (SCLC) [13,22,26,30].

Fowler–Nordheim (F-N) tunneling occurs when the applied electric field is large enough so the electron wave function may penetrate through the triangular potential barrier into the conduction band of the dielectric [31]. 

The expression of the F-N tunneling current is shown by Equation (1).
(1)$$J=\frac{q^{3}E^{2}}{8\pi hq\phi_{B}}exp\left[\frac{-8\pi {\left(2qm_{T}^{*}\right)}^{1/2}}{3hE}\phi_{B}^{3/2}\right]$$
where *J* is the current density, *E* is the applied electric field, *q* is the electronic charge, *h* is Planck‘s constant, $\phi_{B}$ is barrier height for tunneling, $m_{T}^{*}$ is the tunneling effective mass.

The slope of F-N plot can be expressed by Equation (2):(2)$$m=-6.83\times 10^{7}\sqrt{\left(\frac{m_{T}^{*}}{m_{0}}\right)\phi_{B}^{3}}$$

The barrier height can be evaluated from the slope of the linear plot according to Equation (1), as expressed by Equation (3):(3)$$\phi_{B}={\left(\frac{m}{-6.83\times 10^{7}{\left(\frac{m_{T}^{*}}{m_{0}}\right)}^{1/2}}\right)}^{2/3}$$

Figure 7 and Figure 8 show that the dominant conduction mechanism in the region where electroluminescence occurs is the Fowler–Nordheim tunneling for both devices. In this region, ln(J/E^2^) depends linearly on 1/E according to Equation (1). Figure 8 shows a plot of ln(J/E^2^) versus (1/E) of the PIN device and the linear fit in the region where electroluminescence occurs. The barrier height ($\phi_{B}$ = 0.85 eV) was obtained by Equation (3). 

Figure 9a,b shows the plot of ln(J/E^2^) vs. 1/E of the PIN^+^N device under FB and RB, respectively. The barrier height values calculated were 0.6 eV in FB and 0.98 eV in RB. These values indicate that in RB a higher voltage is required so that the injection of carriers into the active layer is possible according to EL and J-E characteristics.

The $\phi_{B}$ of the PIN device was 0.85 eV, while for the PIN^+^N device, it was 0.6 eV, both in FB. This reduction in the height of the barrier was achieved due to the addition of the N-type layer to one of the devices, as shown in Figure 9a,b.

The barrier height depends on the E_opt_ of the layers (N, I and P) [11,29] and also on the substrates used during the processes. For example, in a PIN structure fabricated on a glass substrate with E_opt_ of P, I, and N layers of 2.0 eV, 2.68 eV and 2.0 eV, respectively, a barrier height of 0.25 eV was obtained [11]. The values of the barrier height obtained in this work are higher since we used silicon substrate as the N-layer of the devices. 

Finally, Table 3 summarizes the electroluminescence properties of the devices, as the maximum intensity, the FWHM, the voltage and current density in which the EL starts and the barrier height (ϕ_B_) of devices.

Based on the characteristics of each film (Table 2) and the obtained barrier height values (Table 3), band diagrams were proposed. Figure 9 shows the band diagrams for the PIN and PIN^+^N devices in forward bias (FB) and reverse bias (RB). For the PIN device in FB, shown in Figure 9a, both the P-type a-Si_1−x_C_x:_H, B layer and the N-type c-Si substrate are expected to act as injectors of holes and electrons, respectively. In the I-layer, the electron-hole pair radiative recombination takes place near I/P interface due to the mobility of electrons [22]. Because of the difference of E_opt_ in the different materials, it exists discontinuity of the bands at the I/P (∆E_v_) and N/I (∆E_c_) interfaces, so the holes and electrons injection are limited by the P/I and N/I interface barriers. The discontinuity of bands (∆E_c_) is approximately the barrier height (ϕ_B_), and hole current is considered to be only a minor component because the barrier height for holes to the tunnel (∆E_v_) is larger than that for electrons by a factor of 2 to 3 [13].

The PIN^+^N device is similar to the PIN device, except that the N^+^ type layer was added before the active layer to reduce the barrier height of the carriers in the N/I interface (N-type a-Si:H/I a-Si_1−x_C_x_:H) and increase the carrier injection efficiency to improve the EL intensity. In this device, there is a discontinuity of bands at the I/P (∆E_v_) and N^+^/I (∆E_c_) interfaces, as can be seen in the energy band diagram in Figure 9b. In FB, the electrons of the N-type Si substrate are injected into the N-type a-Si:H layer and then into the conduction band of the I-layer. At the same time, the holes traverse from the P-layer to the localized states near the valence band of the I-layer, where the electron-hole pairs recombination takes place. Figure 9c shows the band diagram for the PIN^+^N device in RB; in this case, a depletion region was created at the N^+^/I and I/P interfaces and higher voltages were needed than in FB to make the injection of electrons possible and holes into the active layer. In these diagrams, solid circles and open circles refer to electrons and holes, respectively. In addition, the tunneling of the carriers to the active layer is illustrated.

## 4. Conclusions

In this work, we reported on the electroluminescence properties of two devices, PIN and PIN^+^N, fabricated on a N-silicon substrate using an active layer of a-Si_1−x_C_x_:H with an optical band gap of 3.1 eV. Both devices exhibited red EL at voltages larger than 20 V over the entire device area of 1 mm^2^, which were seen with the naked eye. The PIN device presented EL only in FB, while the PIN^+^N device did so in FB and RB due to the introduction of an N-type a-Si:H layer between the I-type active layer and the N-type layer. In FB, the PIN^+^N device exhibited a higher carrier injection efficiency, resulting in a higher intensity of EL than that of the PIN device because of the lower barrier height provided by the a-Si:H layer. By the typical plots of ln(J/E^2^) vs. 1/E, we deduced the dominant carrier conduction mechanism in the region where EL takes place is Fowler–Nordheim tunneling. Finally, we demonstrated that it is possible the fabrication of LEDs on a c-Si substrate, fully compatible with the CMOS process, due to the low temperature of fabrication, for applications on monolithic photonics systems, where CMOS transistors, sensors and LEDs can coexist in the same wafer.

## Figures and Tables

**Figure 1 micromachines-13-01948-f001:**
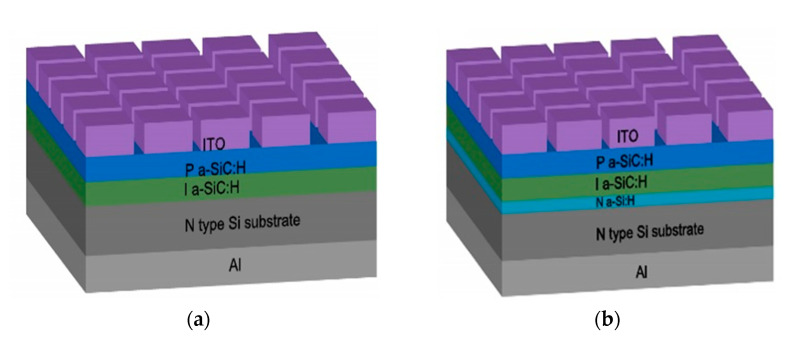
Schematic diagram of the LEDs fabricated: (**a**) PIN, and (**b**) PIN^+^N.

**Figure 2 micromachines-13-01948-f002:**
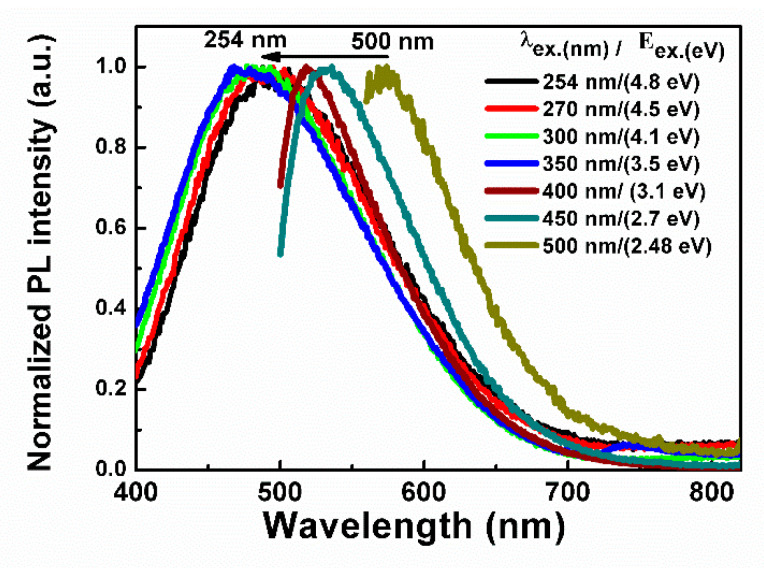
Normalized PL spectra of a-Si1-xCx:H layer with different excitation energy (Eex.).

**Figure 3 micromachines-13-01948-f003:**
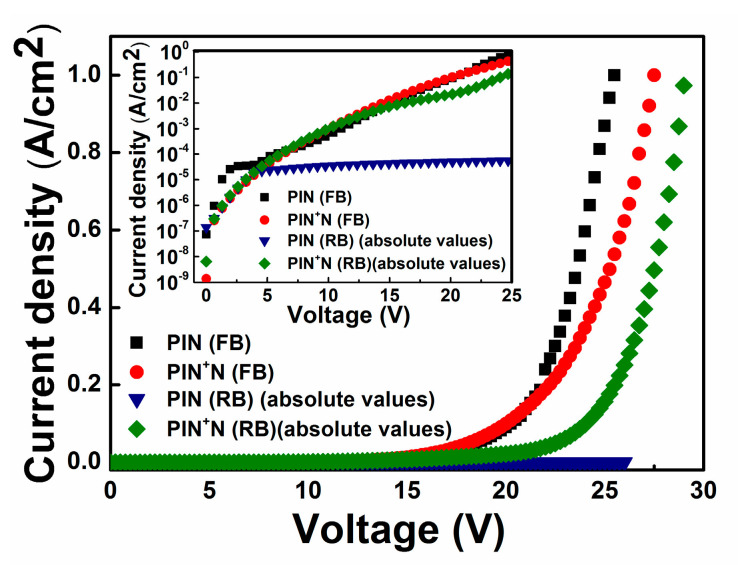
J vs. V characteristics in forward bias (FB) and reverse bias (RB) of the PIN and PIN^+^N devices. The inner graph shows Ln (J) vs. V in forward and reverses bias (absolute voltage values).

**Figure 4 micromachines-13-01948-f004:**
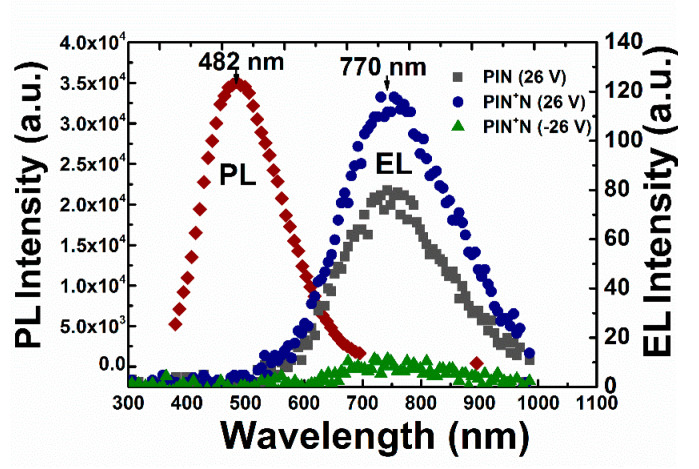
PL intensity spectra of a-Si_1−x_C_x_:H active layer and EL intensity spectra of the PIN (in FB) and PIN^+^N (in FB and RB) devices.

**Figure 5 micromachines-13-01948-f005:**
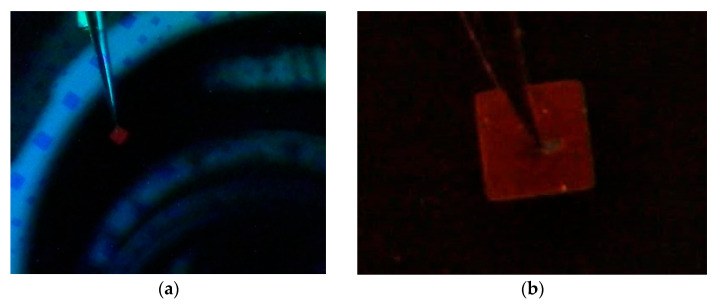
(**a**) PIN^+^N device of 1 mm^2^ emitting red light, (**b**) enlarged image of the same PIN^+^N device.

**Figure 6 micromachines-13-01948-f006:**
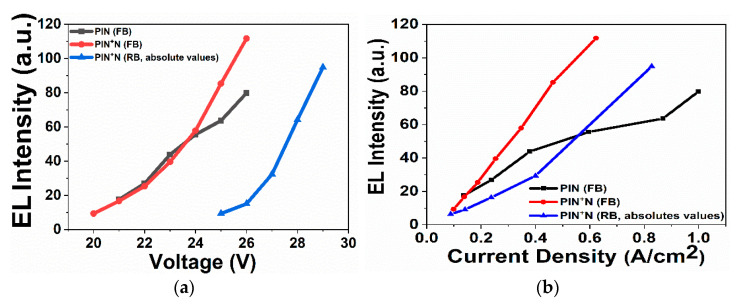
(**a**) EL intensity as a function of voltage and (**b**) EL intensity as a function of the current density of the PIN device in FB and the PIN^+^N device in FB and RB.

**Figure 7 micromachines-13-01948-f007:**
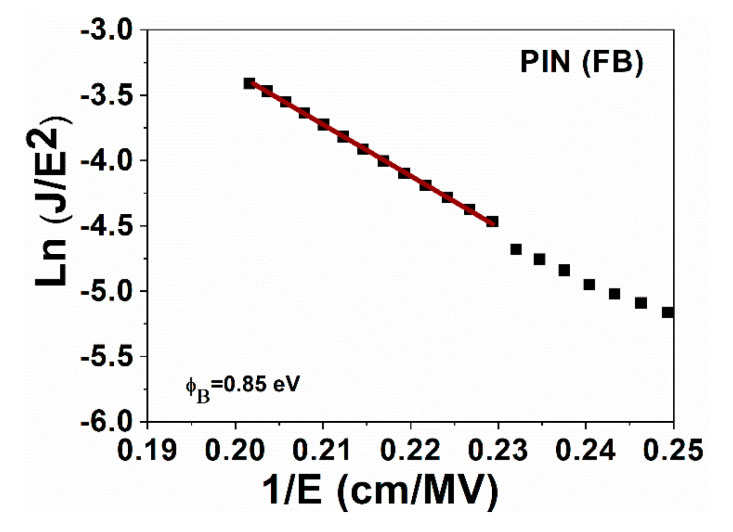
Ln (J/E^2^) vs. 1/E of the device PIN.

**Figure 8 micromachines-13-01948-f008:**
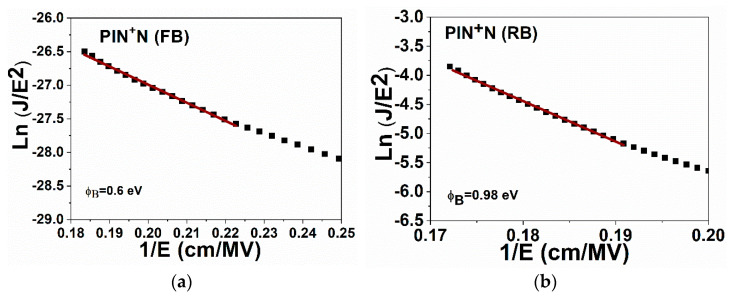
Ln (J/E^2^) vs. 1/E for: (**a**) forward bias and (**b**) reverse bias of device PIN^+^N.

**Figure 9 micromachines-13-01948-f009:**
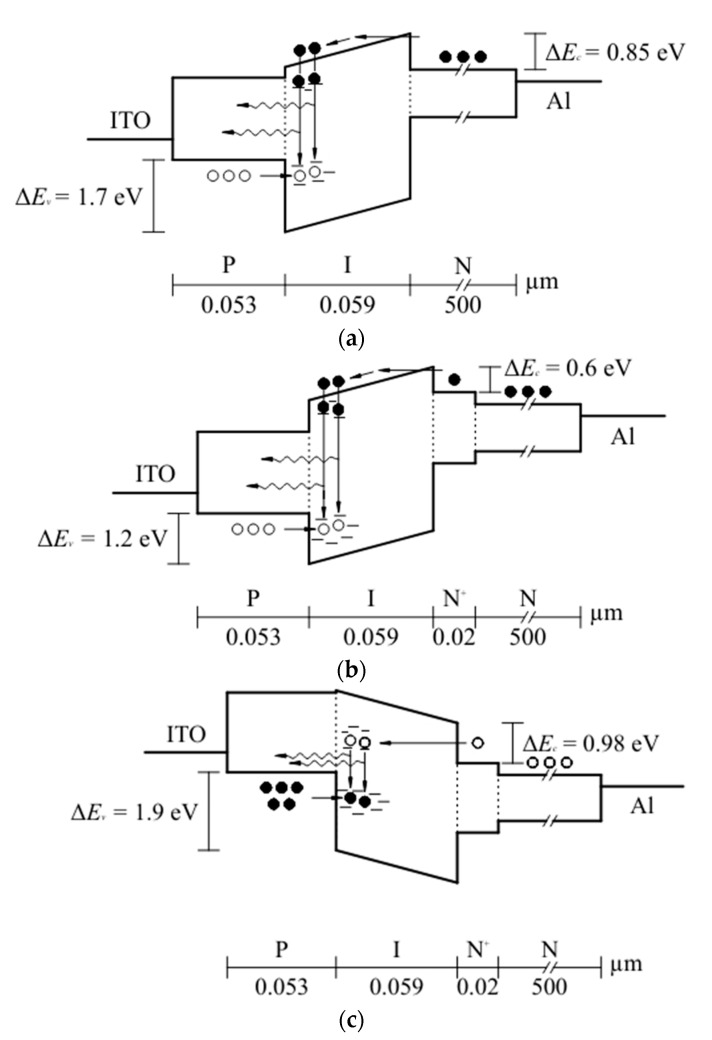
Energy band diagram of the devices: (**a**) PIN in FB, (**b**) PIN^+^N in FB, and (**c**) PIN^+^N in RB.

**Table 1 micromachines-13-01948-t001:** Deposition parameters of N^+^, P and I layers.

Layer		Pressure (Torr)	Power (watts)	SiH_4_ (sccm)	PH_3_ (sccm)	CH_4_ (sccm)	B_2_H_6_ (sccm)
N^+^	a-Si:H	0.55	4	50	2	---	---
I	a-Si_1−x_C_x_	0.7	15	6	---	30	---
P	a-SiC:H	0.55	4	50	---	7.5	5

**Table 2 micromachines-13-01948-t002:** Physical properties of each layer of the PIN and PIN+N structure and chemical composition of the active layer.

Layer	Thickness (µm)	E_opt_ (eV)	Conductivity (S/cm)	Chemical Composition of a-Si_1−x_C_x_:H Layer		
P-a-SiC:H	0.053	1.94	1.39 × 10^−4^	Si (%)	C (%)	O (%)
I-a-Si_1−x_C_x_:H	0.059	3.1	5.27 × 10^−8^	32.4	11.2	56.4
N^+^-a-Si:H	0.020	1.69	1.73 × 10^−4^			
N-Si-n (substrate)	500	1.12	0.3–0.1			

**Table 3 micromachines-13-01948-t003:** Electroluminescence properties of PIN and PIN^+^N devices.

Device	EL (nm)/Maximum Intensity (a.u.)	FWHM of EL (nm)	Voltage of EL (V)	J of EL (A/cm^2^)	ϕ_B_ (eV)
PIN (FB)	765/76.8	240	21	0.09	0.85
PIN^+^N (FB)	762/120.7	220	20	0.13	0.6
PIN^+^N (RB)	770/95	210	−25	0.08	0.98

## Data Availability

Not applicable.

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
