# Peer review of "Development of Heterojunction c-Si/a-Si1−xCx:H PIN Light-Emitting Diodes"

_micromachines, 2022, doi:10.3390/mi13111948_

Round 1

Reviewer 1 Report

Comments

In this manuscript, the authors demonstrated that two different structures of heterojunction LEDs were fabricated at low temperatures by using amorphous silicon-carbide (a-Si1-xCx:H) as an active layer. The PIN device presented EL only in forward bias, while the PIN+N device did it in both, forward and reverse bias. However, some parts of this article are unclear, and some statements I think are incorrect. For example, as follows:

(1) The necessity and innovation of this work cannot be seen from the Abstract.

(2) The introduction is too redundant and gives an incomplete overview of previous research.

(3) The sources and models of the instruments, equipment, and materials involved in this paper are unclear, and it is difficult for others to repeat the experimental process.

(4) In Figure 2, the authors declared that we can see that PL spectra shift to lower wavelength from red to blue by reducing the excitation energy (Eex.) because of activation of different recombination centers. However, what I have observed is: as the excitation energy decreases, the fluorescence emission spectrum is blue-shifted and then red-shifted. This is a false statement.

(5) In Figure 4. The authors declared that there is a large Stokes shift of 288 nm between EL and PL spectra. However, the fluorescence spectrum is red-shifted from the corresponding absorption spectrum, which is called the Stokes shift. 

Author Response

Dear reviewer,

Thank you for your revision and comments, because that improves the quality of the manuscript.

Here are our answers to your comments:

(1) The necessity and innovation of this work cannot be seen from the Abstract.

Authors Response:

Taking into account your comments, we agree to stress the innovation of our work, and we have modified the abstract to stress that innovation.

(2) The introduction is too redundant and gives an incomplete overview of previous research.

Authors Response:

We agree, we have improved the introduction section in order to give an overview of previous work and also to be more clear in our ideas.

(3) The sources and models of the instruments, equipment, and materials involved in this paper are unclear, and it is difficult for others to repeat the experimental process.

Authors Response:

We have included models and brands of the equipment used 

(4) In Figure 2, the authors declared that we can see that PL spectra shift to lower wavelength from red to blue by reducing the excitation energy (Eex.) because of activation of different recombination centers. However, what I have observed is: as the excitation energy decreases, the fluorescence emission spectrum is blue-shifted and then red-shifted. This is a false statement.

Authors Response:

We agree, we have made the corrections in the manuscript, and thank you for this observation.

(5) In Figure 4. The authors declared that there is a large Stokes shift of 288 nm between EL and PL spectra. However, the fluorescence spectrum is red-shifted from the corresponding absorption spectrum, which is called the Stokes shift. 

Authors Response:

We agree, we have made the corrections in the manuscript, and once again thank you for this observation.

Reviewer 2 Report

1. Hole-only and electron-only devices for devices with two different device architectures might give some information on the difference in their carrier transporting ability.

2. As shown in Figure 4, there is a large difference between PL maximum and EL maximum. An explanation for that should be given.

3. The full name of CMOS should be given.

Author Response

Dear reviewer,

Thank you for your revision and comments, because that improves the quality of the manuscript.

Here are our answers to your comments:

  1. Hole-only and electron-only devices for devices with two different device architectures might give some information on the difference in their carrier transporting ability.

Authors Response:

The recombination of holes and electrons can be of two types, radiative and non-radiative, for PIN devices is very important to have both kind of carriers in order to have radiative recombination, where emission of photons is produced. Most of radiative occurs in defects of different kinds in the amorphous active layer. The PIN+N device is similar to PIN device, except that a N+ type layer was added before the active layer to reduce the barrier height of the carriers in the N/I interface (N-type a-Si:H/I a-Si1-xCx:H) and increase the carrier injection efficiency to improve the EL intensity.

  1. As shown in Figure 4, there is a large difference between PL maximum and EL maximum. An explanation for that should be given.

Authors Response:

We have modified the text in order to be more clear in the explanation:

The shift between PL and EL can be caused by distortion of lattices, the relaxation of carriers into deep gap states before the recombination takes place [3] and/or by the excitation of carriers by the electric field due to impact ionization in the localized states and in the extended states of the valence and conduction bands, resulting in the lower energy shift and broadening of the EL spectra compared to the PL spectra [10].

  1. The full name of CMOS should be given.

Authors Response:

We agree, we have made the correction in the manuscript. Complementary Metal-Oxide-Semiconductor (CMOS).

Reviewer 3 Report

It is recommendable to ammend the next items:

1. Row 187 said "layer with higher energy level, therefor in the PIN+N structure"

Please correct to "therefore"

2. Row 207 said: "And the barrier height can be evaluated from the slop of the linear plot according to"

It should be "slope"

3. Row 230 said: "this work are higher since we used silicon substrate as N-layer as active part of the devices.

The "active-layer" is considered and utilized for recombination or photo emission. Even though, N-layer, is considered an electric contact, but not be considered an active layer.

4. Fig. 9 (Band diagram of the structures)

Please, correct and rewrite the band diagrams considering the vertical magnitude (eV).

i.e. the corresponding Delta of Ec or Ev values, are not proportional to the bandgap values. See Table 2, its Eopt. values

Author Response

Dear reviewer,

Thank you for your revision and comments, because that improves the quality of the manuscript.

Here are our answers to your comments:

  1. Row 187 said "layer with higher energy level, therefor in the PIN+N structure"

Please correct to "therefore"

Authors Response:

We thank the reviewer for this observation, which has been corrected in the manuscript.

  1. Row 207 said: "And the barrier height can be evaluated from the slop of the linear plot according to" It should be "slope"

Authors Response:

We thank again the reviewer for this observation, which has been corrected in the manuscript.

  1. Row 230 said: "this work are higher since we used silicon substrate as N-layer as active part of the devices.

The "active-layer" is considered and utilized for recombination or photo emission. Even though, N-layer, is considered an electric contact, but not be considered an active layer.

Authors Response:

We agree, we have made the correction in the manuscript.

  1. Fig. 9 (Band diagram of the structures)

Please, correct and rewrite the band diagrams considering the vertical magnitude (eV).

i.e. the corresponding Delta of Ec or Ev values, are not proportional to the bandgap values. See Table 2, its Eopt. values

Authors Response:

We agree, we have modified Figure 9 as was suggested by the reviewer.
